# Experimental Characterization of Laser Trepanned Microholes in Superalloy GH4220 with Water-Based Assistance

**DOI:** 10.3390/mi13122249

**Published:** 2022-12-17

**Authors:** Liang Wang, Huayu Yang, Naifei Ren, Zhengtian Wu, Kaibo Xia

**Affiliations:** 1Faculty of Mechanical and Materials Engineering, Huaiyin Institute of Technology, Huaian 223000, China; 2School of Mechanical Engineering, Jiangsu University, Zhenjiang 212013, China; 3School of Electronic and Information Engineering, Suzhou University of Science and Technology, Suzhou 215000, China

**Keywords:** laser trepanning, water-based assistance, superalloy, millisecond laser

## Abstract

An experiment using water-assisted millisecond laser trepanning on superalloy GH4220 was carried out, and the effects of pulse energy on the hole entrance morphology, diameter, roundness, cross-section morphology, taper angle, sidewall roughness, and recast layer in air and with water-based assistance were compared and analyzed. The results show that, compared with the air condition, the water-based assistance improved the material removal rate and hole quality, increased the diameter of the hole entrance and exit, increased the hole roundness, decreased the hole taper angle, decreased the hole sidewall roughness, and reduced the recast layer thickness. In addition, under the combined action of water and steam inside the hole, the sidewall surface morphology quality was improved. Compared with the air condition, the spatter around the hole entrance was reduced, but the oxidation phenomenon formed by the thermal effect surrounding the hole entrance with water-based assistance was more obvious. The research provided technical support for the industrial application of millisecond laser drilling.

## 1. Introduction

Laser drilling has the advantages of no tool loss, high machining precision, low cost, high drilling efficiency, etc. It can be used to drill high-aspect ratio holes in a wide variety of materials. Nowadays, laser drilling has been widely used in aerospace, aircraft, medical devices, automobile industries, etc. [1,2,3,4,5,6]. Millisecond laser drilling is widely used in industrial applications. However, some defects, such as microcracks, heat-affected zones, and recast layers, are generated during millisecond laser drilling [7,8,9,10].

Scholars have found that water assistance could improve the quality of laser drilling. Zhu et al. [11] improved the quality of picosecond laser drilling using the water-assisted method on the back of the workpiece, and a large number of holes with an exit diameter of 55 μm were drilled on a 60-μm-thick stainless steel sheet. After the sheet was drilled through, water entered into the hole through the capillary phenomenon, and the laser reflected at the gas–liquid interface. After the water medium was radiated, the mechanical effect and cavitation effect would be generated. Combined with the cooling effect of water, the quality of the hole was improved. At the same time, the water-assisted method on the back of the workpiece could also reduce the hole taper angle, the recast layer, and the heat-affected zone generated in the laser drilling. Behera et al. [12] introduced water-assisted laser drilling using different pulse durations (millisecond, nanosecond, and femtosecond) and compared it with the traditional laser drilling method, proving the superiority of water-assisted laser drilling, and the shorter the pulse duration, the better the drilling quality. They found that the mechanism of water-assisted laser drilling mainly includes light transmission (absorption) in liquid, liquid heating and vaporization, bubble evolution (formation, growth, and collapse) and material evolution (heating, melting, and vaporization). Chen et al. [13] conducted a nanosecond laser drilling of silicon nitride ceramics underwater and compared it with the air environment. It was found that the underwater environment had a more obvious effect on the hole taper. When the scanning speed was constant, the deeper the hole, the smaller the taper. Under the same hole depth, the taper decreased with the decrease in the scanning speed. At the same time, underwater laser drilling could obtain a better hole sidewall. Feng et al. [14] carried out picosecond laser drilling on zirconia underwater to study the change in the hole geometric quality after optimizing the process parameters. The experimental result showed that the cracks on the hole sidewall were reduced, and the hole sidewall surface roughness was reduced in the water environment. Wang et al. [15] selected a single-crystal silicon wafer (4H-SiC) with a thickness of 500 μm as the experiment material to study the effect of water assistance on femtosecond laser drilling. It was found that phenomena such as inlet debris redeposition, cracks, surface material falling off, heat-affected zones, and recast layers could be eliminated by water scouring and diffusion in the hole. In addition, the water-layer thickness and the pulse repetition rate had a great impact on the drilling efficiency and hole quality.

Above all, although research on water-assisted laser drilling has been carried out at present, there is a lack of systematic research and analysis on water-assisted laser drilling, and there were no relevant reports on water-assisted millisecond laser trepanning in superalloy GH4220. In order to further improve the quality of millisecond laser drilling, the water-based assisted laser trepanning method was used to carry out laser drilling experiments. The effects of air and water-based assistance on the hole morphology, diameter, roundness, cross-sectional morphology, taper angle, sidewall roughness, and recast layer thickness were compared and analyzed. This research could provide technical support for the industrial application of millisecond laser drilling.

## 2. Materials and Methods

The water-assisted laser trepanning device is shown in Figure 1. It mainly included a machining head, workpiece, tank, fixture, and motion platform. The laser used was an Nd: YAG laser; the laser parameters are shown in reference [16]. In millisecond laser trepanning, the beam was applied perpendicularly to the workpiece, and the laser focus was located on the upper surface of the workpiece. The relative circular motion between the laser beam and the workpiece surface was formed by controlling the movement trajectory of the workpiece on the horizontal plane, and the diameter of the shown cutting path was 400 μm, as shown in Figure 1. The upper surface of the workpiece was exposed to air, and the lower surface was submerged in water, with a submerged depth of about 1.6 mm (the workpiece thickness). In order to reduce the influence of the assist gas on the water, the lower air pressure allowed by the equipment was used. The assist gas used was argon with a pressure of 0.1 MPa (the assist gas was applied by coaxial blowing). The parameters used are shown in Table 1; the number of circles indicates how many circles the workpiece moved.

The nickel-based superalloy GH4220 (Manufacturer name: Dongguan Tengfeng Metal Materials Co., Ltd.; Dongguan, China) used in this experiment needed to be prepared before the laser drilling. The purchased superalloy bar (with a diameter of 30 mm) was cut into sheets of the same thickness by wire cutting, and a certain polishing allowance (with a polishing allowance of 0.1 mm) was reserved. Detergent was used to remove the oil stain remaining on the workpiece surface during wire cutting in order to facilitate subsequent polishing. Afterward, the scratches left by the wire cutting were removed using the metallographic sander and water abrasive paper, the polished workpiece was placed into the beaker and cleaned with an ultrasonic cleaner for 5 min (the cleaning solution was absolute ethanol), and the debris and dirt generated during the grinding process on the workpiece surface were removed. The changes to the workpiece at various stages are shown in Figure 2. The thickness of the workpiece used in the experiment after pretreatment was 1.6 mm.

After the experiment, the workpiece needed to be processed in multiple steps to obtain more experimental data. Grinding was used to remove the spatter around the hole entrance and exit, but in order to prevent the molten material from blocking the hole when grinding the hole cross-section, a large amount of clean water could be added. After grinding, the hole sidewall must be polished with a metallographic polisher to prepare for the subsequent corrosion of the hole cross-section. In this experiment, in order to reduce the uncertainty and measurement error, each group of experiments was repeated three times, and the average value was taken for analysis.

After the experiment, the hole diameter was measured several times from different angles, and the average value was obtained to reduce the measurement error. The angle difference between adjacent diameters measured each time was 30°, as shown in Figure 3. The formula for calculating the hole diameter is shown in Equation (1).
d = (d_1_ + d_6_)/6(1)

The hole’s roundness was described by the hole circularity deviation (the deviation of the hole’s maximum radius and minimum radius). The smaller the circularity deviation, the better the hole’s roundness. The hole circularity deviation (Δr) was calculated using the following Equation (2).
Δr = r_max_ − r_min_(2)

The calculation method for the through-hole taper angle is shown in the reference; the formula was as follows [17]:α = 2arctan((d_1_ − d_2_)/2h) × 180°/π(3)
where α is the taper angle, and d_1_, d_2_, and h are the entrance diameter, exit diameter, and workpiece thickness, respectively.

The hole sidewall roughness (Sa) was measured using KEYENCE confocal laser scanning microscopy (CLSM) analysis software (MultiFileAnalyzer 1.3.1.120).

## 3. Results and Discussion

### 3.1. Spatter around the Hole Entrance

The spatter around the hole entrance in air and with water-based assistance is shown in Figure 4. It was found that the spatter around the hole entrance with water-based assistance was less than that in the air, which indicated that the recoil pressure generated by the water entering the hole after the hole was drilled through enhanced the removal of molten material and debris. In addition, both in the case of air and water-based assistance, there was a thermal effect area around the hole (oxidation area). This oxidation area deviated from the side of the hole. It was mainly caused by the laser trepanning method used in this experiment. With the movement of the laser beam, a blind hole was first formed in the action area during the initial period of the laser beam movement. With the continued movement of the laser beam, under the action of assist gas, plasma and spatter were removed from the formed blind hole. Because the previously formed blind hole was located near the laser beam motion path and deviated from the center of the final hole, more molten material, spatter, and plasma were removed from one side (the position of the blind hole formed during the initial period of laser beam movement) of the hole entrance, resulting in a more obvious thermal effect on this side.

Furthermore, it was found that the oxidation phenomenon formed by the thermal effect surrounding the hole entrance with water-based assistance was more obvious. It was because the water flowed into the hole after the workpiece was drilled through. The water medium promoted the removal of the material. Part of the water was heated and evaporated under the action of the laser. The water mixed with melt and material debris was removed from the hole under the power of the steam. The material removal process was more intense than that in the air, resulting in a more obvious thermal effect.

With the increase in the pulse energy, the spatter around the hole entrance in air and with water-based assistance decreased. With the increase in the pulse energy, the material was easier to drill through, and the molten material and debris could be ejected from the hole exit in time along with the assist gas. At the same time, because the hole exit diameter increased with the increase in pulse energy, more molten material and debris could pass through and be removed, and the spatter around the hole entrance was reduced significantly. Due to the effect of the assist gas, the spatter accumulation around the hole exit increased; as it was difficult to photograph, the spatter around the hole exit was not analyzed.

### 3.2. Hole Entrance/Exit Diameter, Roundness

Figure 5 shows the morphology of the hole entrance and exit after grinding. The effect of water-based assistance on the hole morphology was not obvious, which was mainly because the water had little influence on the laser drilling process (there was no water above the sample before laser drilling) before the hole was drilled through. After the hole was drilled through, some water entered the hole, which promoted the removal of molten materials and ultimately led to an increase in material removal. However, due to the limited increase in material removal, the observed change in hole morphology was not obvious. In addition, due to the influence of the assist gas, the promotion effect of the water medium was reduced to a certain extent.

Figure 6 shows the effect of pulse energy on the hole entrance and exit diameter. With the increase in pulse energy, the hole entrance and exit diameter increased. The reason was that with the increase in pulse energy, the workpiece would absorb more energy, and the material removal rate would be improved. In the case of water-based assistance, the hole entrance and exit diameter were slightly larger than that in the case of air. If the workpiece was drilled through, the water entered into the hole from the hole exit; then, the water would be heated and evaporated to steam under the action of the laser. The steam mixed with water droplets generated impact force, which promoted the removal of molten material and debris from the hole, thus improving the material removal rate. At the same time, under the action of the laser, bubbles would also be generated in the water, and the bubbles would rise and break to generate shock waves and micro jet, promoting the removal of molten material and debris from the hole and further improving the material removal rate. In addition, the water suppressed the expansion of plasma during the drilling, the plasma shielding effect was weakened, and more material was removed.

Figure 7 shows the hole entrance and exit circularity deviation in air and with water-based assistance. It was found that the hole entrance and exit circularity deviation with water-based assistance was smaller than that in the case of air. This was because after the workpiece was drilled through, the water promoted the removal of molten material and debris, suppressed the expansion of plasma during drilling, and caused the absorption of laser energy by the hole’s sidewall to be more uniform [12,17,18,19]. Therefore, hole roundness was better than that in the case of air. With the increase in pulse energy, the change in the hole’s entrance and exit circularity deviation under these two conditions was not obvious.

### 3.3. Hole Cross-Section Morphology, Taper Angle

The cross-sectional morphology of the hole in the case of air and with water-based assistance is shown in Figure 8. The calculated taper angle is shown in Figure 9. It could be found that with the increase in pulse energy, the removal rate of the material was improved. With the increase in pulse energy, the taper angle decreased continuously, but when the pulse energy increased to a certain range, the taper angle tended to become larger. This was because, with the increase in pulse energy, the hole would be drilled through faster, and a large amount of molten material could be removed from the hole in time. However, when the laser energy increased to a certain range, a large amount of energy accumulated on the upper surface of the workpiece, and the material removal rate at the hole entrance increased obviously, resulting in a larger taper angle. When the pulse energy was 1.9 J, the hole taper angle was larger than 1.6 J.

### 3.4. Hole Sidewall Morphology and Roughness

Figure 10 is the 2D morphology of the hole sidewall near the hole entrance, middle, and exit in air and with water-based assistance, and the pulse energy is 0.7 J. It was found that the molten material on the hole sidewall in the air condition was in the form of liquid droplets, and the number was also large. The morphology of the hole sidewall with water-based assistance was relatively smooth.

Figure 11 is the 3D morphology corresponding to Figure 10. It was found that the hole sidewall roughness with water-based assistance was less than that in the air condition. When the pulse energy was 0.7 J, the hole sidewall roughness at the hole entrance, middle, and exit with water-based assistance was reduced by 41%, 10%, and 19%, respectively, compared with that in the case of air. When the laser irradiated the water, the steam generated by the heating of the water continuously washed the hole sidewall, making the residual molten material distribution on the sidewall more uniform. At the same time, after the workpiece was drilled through, bubbles were generated under the action of the laser, and the sidewall material was removed more evenly due to the impact force and micro jet caused by the bubbles [17]. As a result, more material was removed from the hole, and the hole sidewall quality was improved. The hole roughness decreased from the entrance to the exit. With the increase in hole depth, the laser energy was continuously consumed, and the energy absorbed by the material near the hole exit was reduced, which led to a decrease in the material removal rate near the hole exit and the uneven removal of the sidewall material. After the hole was drilled through, under the action of assist gas and gravity, more molten material and debris would be ejected from the hole exit, resulting in more molten material remaining on the sidewall near the hole exit. Therefore, the hole sidewall roughness near the hole exit was larger than that of other parts, both in the case of air and with water-based assistance.

Figure 12 shows the effect of pulse energy on the sidewall roughness of the different parts of the hole. It was found that with the increase in pulse energy, the roughness of each part decreased and, finally, stabilized within a certain value range. This was maybe that when the laser energy increased, the material removal on the hole sidewall was more uniform. When the pulse energy reached a certain range, the influence of the pulse energy change on the sidewall roughness would be reduced. When the laser pulse energy was 1.3–1.9 J, the sidewall roughness had little change. The water-based assistance could reduce the hole sidewall roughness and improve the hole sidewall quality.

### 3.5. Recast Layer

Figure 13 shows the thickness of the recast layer at the hole entrance, middle, and exit in air and with water-based assistance, and the pulse energy was 0.7 J. Figure 14 shows the change in the recast layer thickness at the hole entrance, middle, and exit under different pulse energies and different environments (in air and with water-based assistance). It was found that the recast layer thickness (at the hole entrance, middle, and exit) of the hole drilled with water-based assistance was smaller than that drilled in air. When the pulse energy was 0.7 J, the recast layer thickness at the hole entrance, middle, and exit with water-based assistance was reduced by 32%, 17%, and 23%, respectively, compared with that in air. The reason was that after the workpiece was drilled through, under the combined action of water and steam (generated by heating of the water), the molten material and spatter could be effectively removed, and the residual molten material on the hole sidewall was reduced, which led to the reduction in the recast layer thickness [17]. In addition, it was also found that the distribution of the recast layer near the hole middle was more uniform with water-based assistance.

It was also found, as shown in Figure 14, that when the single pulse energy was 0.7 J, the recast layer thickness on the sidewall at the hole entrance was large in these two environments. This was because when the pulse energy was low, the workpiece would be drilled through for a long time. When the workpiece was not drilled through, a large amount of molten material was ejected from the hole entrance under the combined action of the assist gas and the recoil pressure generated by the evaporation of molten material, resulting in a large amount of molten material remaining on the hole sidewall at the hole entrance.

## 4. Conclusions

The experimental research of using millisecond laser trepanning on a superalloy GH4220 with water-based assistance was carried out, and the effects of different pulse energy on the spatter, hole diameter, roundness, taper angle, sidewall morphology, and roughness, and the distribution of the recast layer in air and with water-based assistance, were studied. This research has industrial application prospects. The main research results were as follows:(1)With the increase in pulse energy, the spatter around the hole entrance and exit decreased in air and with water-based assistance. The spatter around the hole entrance with water-based assistance was less than that in the case of air.(2)With the increase in pulse energy, the hole entrance and exit diameter increased because the material removal rate was increased with the increase in pulse energy. Under the water-based assistance condition, the hole roundness was better than that under the air condition. This was because after the workpiece was drilled through, the water promoted the removal of molten material and suppressed the shielding effect of plasma during laser drilling. Moreover, the absorption of laser energy by the material on the hole sidewall was more uniform, resulting in better roundness than that under the air condition. With the increase in pulse energy, the roundness of the hole entrance and exit in these two environments was not changed significantly.(3)With the increase in pulse energy, the material removal rate increased, and the hole taper angle decreased. However, when the pulse energy increased to a certain range, the hole taper angle tended to become larger, and the hole taper angle at 1.9 J was larger than 1.6 J. Water−based assistance could reduce the hole taper angle.(4)The hole sidewall morphology in the water-based assistance conditions was better than that in the case of air, and the hole sidewall roughness was less than that in air. When the pulse energy was 0.7 J, the hole sidewall roughness at the hole entrance, middle, and exit with water-based assistance was reduced by 41%, 10%, and 19%, respectively, compared with that in air. The sidewall roughness at the hole exit was larger than that of other parts.(5)The recast layer thickness of the hole drilled with water-based assistance was smaller than that in the case of air. This was mainly because after the workpiece was drilled through, the molten material produced during laser drilling could be effectively removed under the combined action of the water and steam, the residual amount of molten material on the hole sidewall was reduced, and the thickness of the recast layer was reduced. When the pulse energy was 0.7 J, the recast layer thickness at the hole entrance, middle, and exit with water-based assistance was reduced by 32%, 17%, and 23%, respectively, compared with that in air.(6)Compared with the air condition, water-based assistance could improve the material removal rate and the hole quality. The hole entrance and exit diameter increased, the roundness increased, the taper angle decreased, the hole sidewall roughness decreased, and the recast layer thickness decreased. At the same time, the hole sidewall surface morphology was better with water-based assistance. Compared with the air condition, the spatter around the hole entrance was reduced, but the oxidation phenomenon formed by the thermal effect surrounding the hole entrance with water-based assistance was more obvious.

## Figures and Tables

**Figure 1 micromachines-13-02249-f001:**
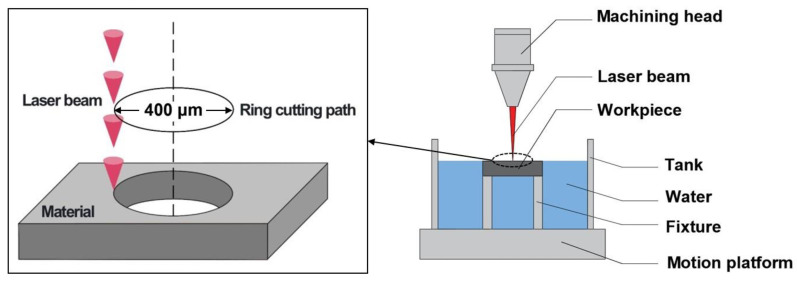
Schematic diagram of millisecond laser trepanning with water-based assistance.

**Figure 2 micromachines-13-02249-f002:**
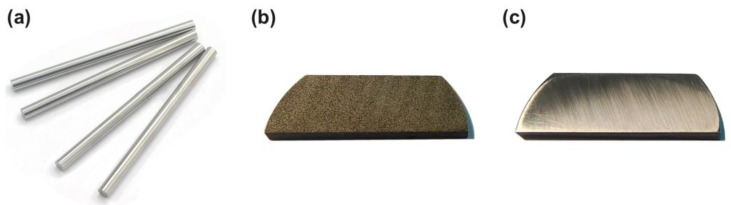
Experimental materials: (**a**) bar, (**b**) sheet (after wire cutting), (**c**) sheet (after grinding).

**Figure 3 micromachines-13-02249-f003:**
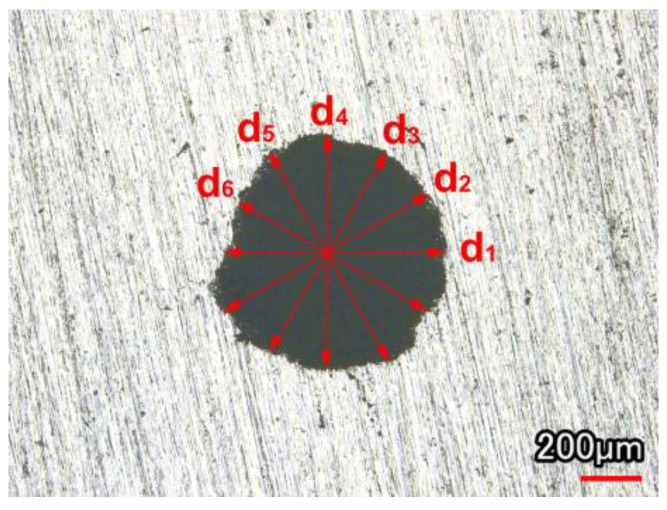
Schematic diagram of diameter measurement.

**Figure 4 micromachines-13-02249-f004:**
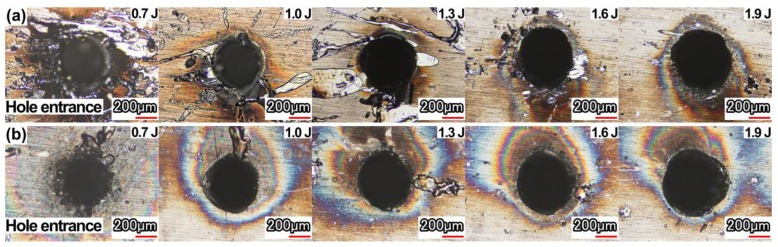
Effect of pulse energy on spatter around the hole entrance: (**a**) in air, (**b**) water-based assistance.

**Figure 5 micromachines-13-02249-f005:**
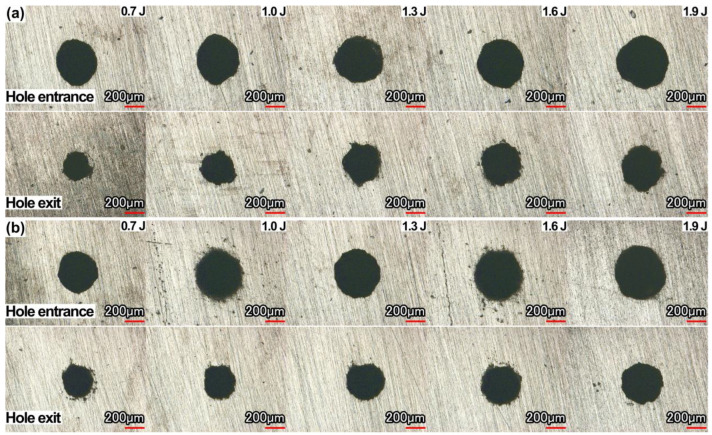
Effect of pulse energy on the morphology of hole entrance and exit: (**a**) in air, (**b**) water-based assistance.

**Figure 6 micromachines-13-02249-f006:**
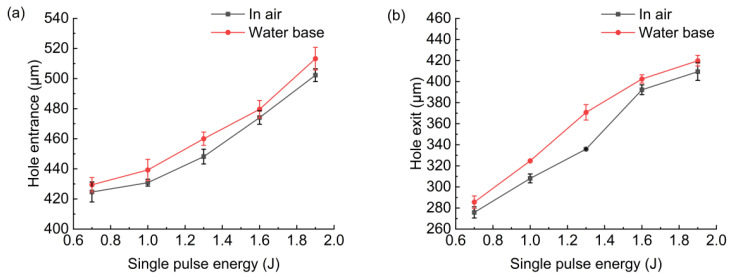
Effect of pulse energy on hole diameters: (**a**) hole entrance, (**b**) hole exit.

**Figure 7 micromachines-13-02249-f007:**
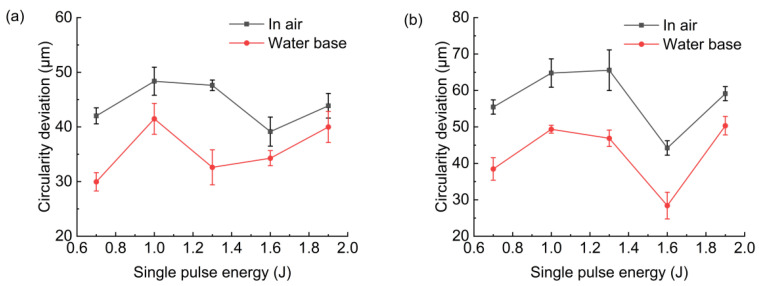
Effect of pulse energy on hole circularity deviation: (**a**) hole entrance, (**b**) hole exit.

**Figure 8 micromachines-13-02249-f008:**
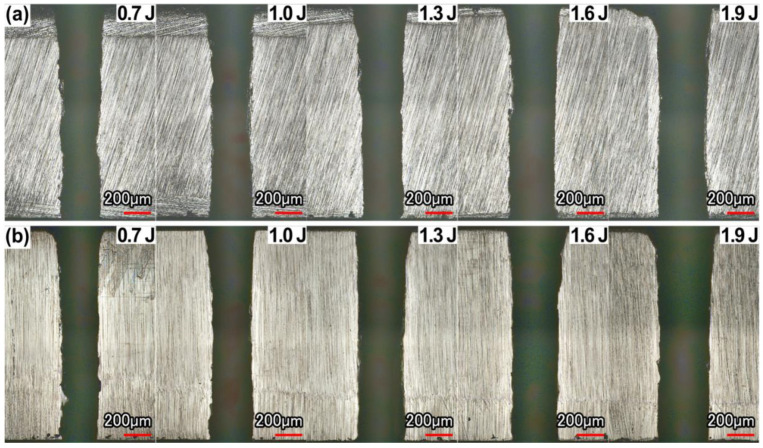
Effect of pulse energy on the morphology of hole cross-section: (**a**) in air, (**b**) water-based.

**Figure 9 micromachines-13-02249-f009:**
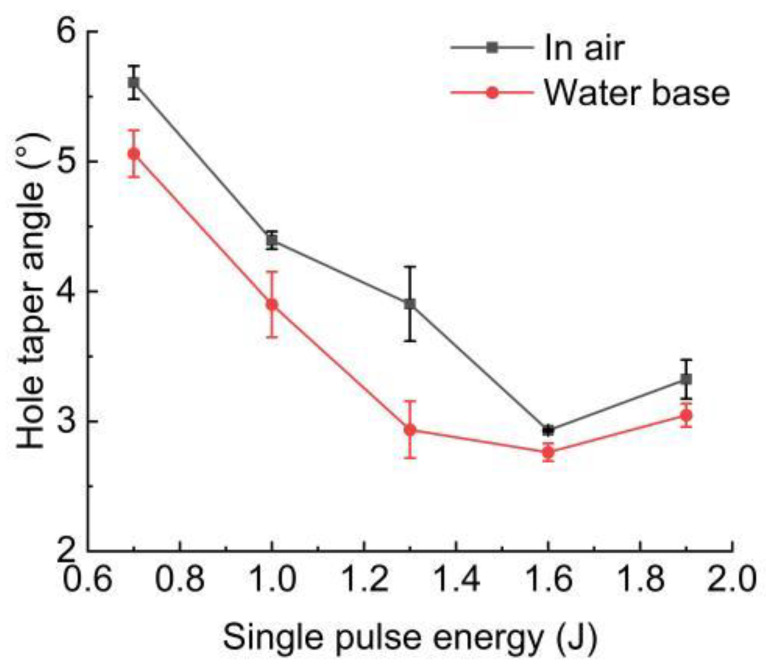
Effect of pulse energy on hole taper angle.

**Figure 10 micromachines-13-02249-f010:**
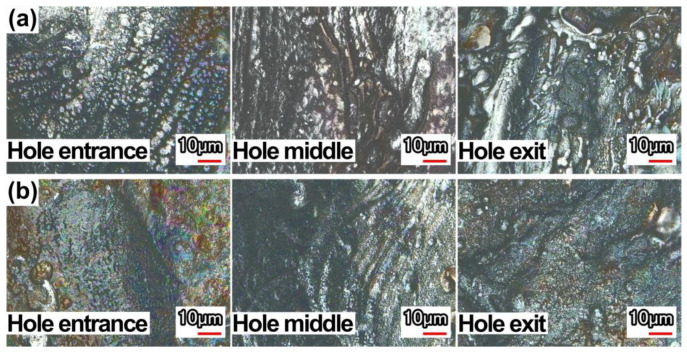
2D morphology at different locations of hole sidewall: (**a**) in air, (**b**) water-based assistance.

**Figure 11 micromachines-13-02249-f011:**
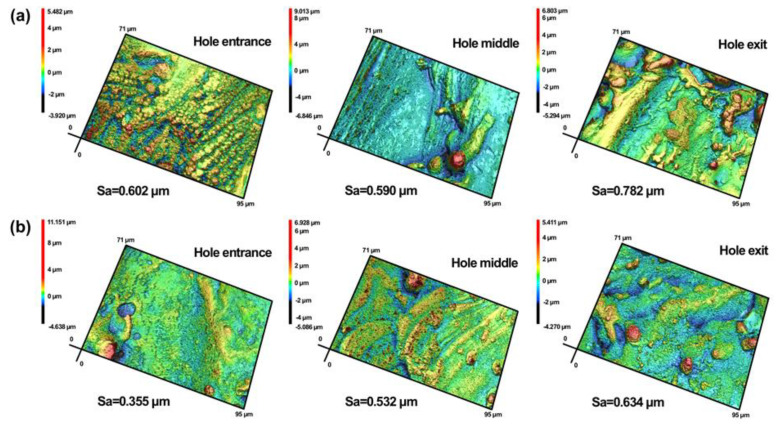
3D morphology at different locations of hole sidewall: (**a**) in air, (**b**) water−based assistance.

**Figure 12 micromachines-13-02249-f012:**
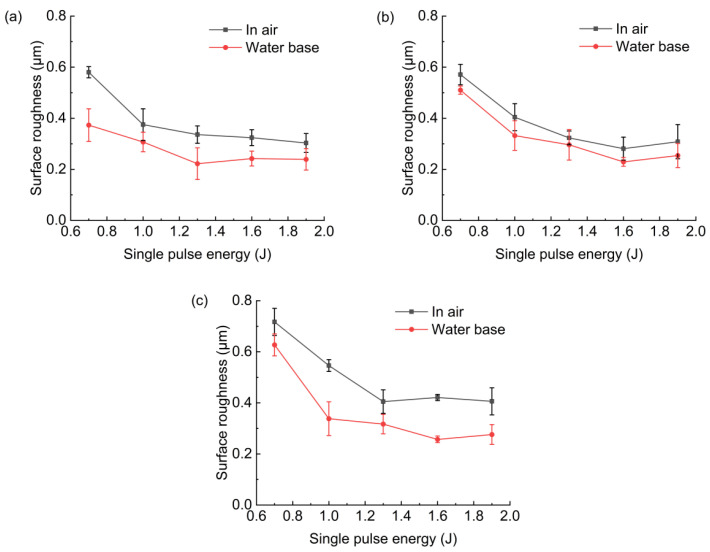
Effect of pulse energy on hole roughness: (**a**) hole entrance, (**b**) hole middle, (**c**) hole exit.

**Figure 13 micromachines-13-02249-f013:**
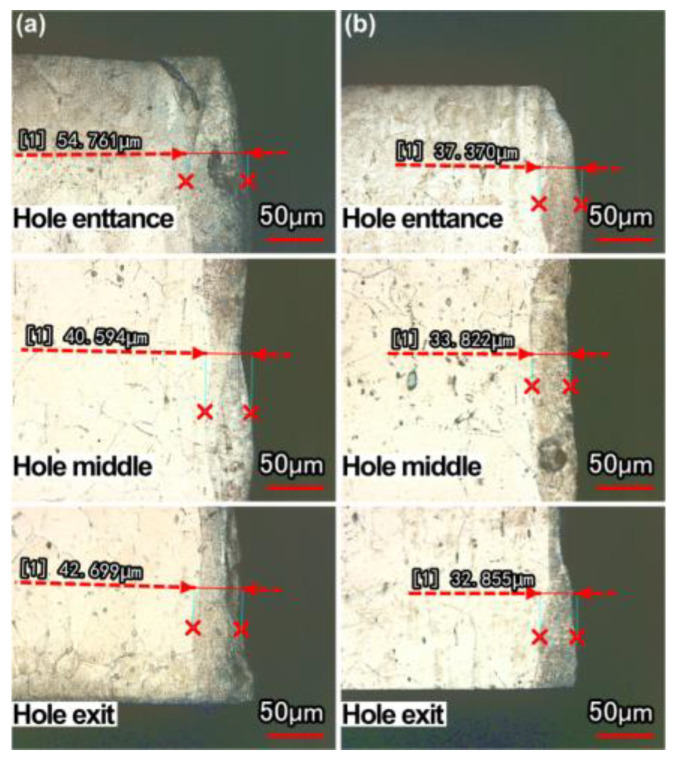
Thickness of recast layer at different locations of hole sidewall: (**a**) in air, (**b**) water−based assistance.

**Figure 14 micromachines-13-02249-f014:**
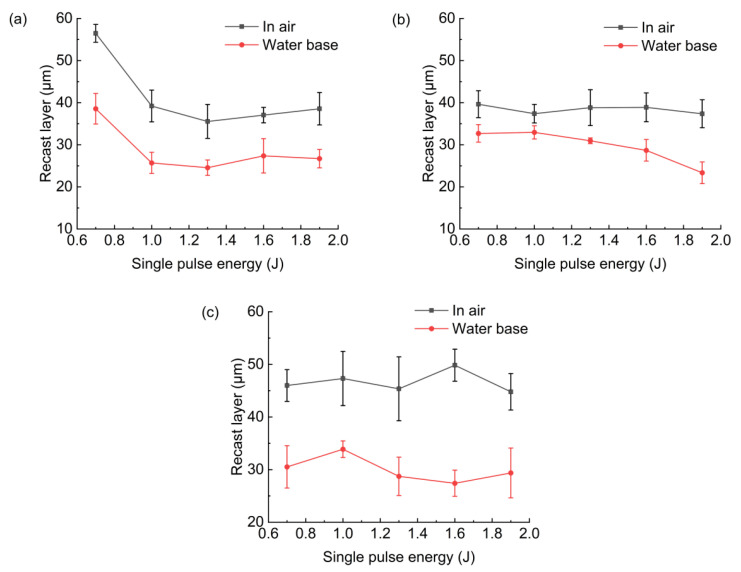
Effect of pulse energy on hole recast layer: (**a**) hole entrance, (**b**) hole middle, (**c**) hole exit.

**Table 1 micromachines-13-02249-t001:** Laser parameters used in air and water-based assistance.

Case	Pulse Duration (ms)	Pulse Repetition Rate (Hz)	Pulse Energy (J)	Number of Circles	Scanning Speed (mm/min)	Environment
I	0.8	60	0.7–1.9	2	50	In air
II	0.8	60	0.7–1.9	2	50	Water-based assistance

## Data Availability

The data supporting the findings of this work are available from the corresponding author upon reasonable request.

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
