# Peer review of "Experimental Characterization of Laser Trepanned Microholes in Superalloy GH4220 with Water-Based Assistance"

_micromachines, 2022, doi:10.3390/mi13122249_

Round 1
Reviewer 1 Report
The manuscript covers fruitful experimental characterization. However, it much tends to be an experiment report rather than a research article. It is not involved the significant academic mechanism discussion. Amendment on it is required if considering of publication in the journal “Micromachines”. How the water to play a role to the laser thermal heat. It is also related the flowing water or static water. Next, the submerged depth is another important factor which is correlated to the metal thickness to be drilled. What ratio of submerged depth to thickness is the optimum, i.e., the optimal submerged depth. Too small a submerged depth may not be efficient; depth equals to thickness or even over the thickness? As an academic study work, these are not clear, it may not provide effective guidance to practical application.
Furthermore, the manuscript is entitled “Microstructural study of laser trepanning in superalloy GH4220 with water-based assistance”, however, don’t see microstructure characterization in the study, such as microstructure phase, grain size and its shape comparison between air and water. In my option, the title may need to revise to precisely fit the experimental study addressed in this manuscript.
Some details:
1) Hole diameter to be drilled needs to indicate in Figure 1.
2) The coloration from thermal effect in Figure 4, both air and water show the orientation instead of a uniform thermal effect surrounding the hole entrance, proper discussion on this phenomenon is required.
3) No obvious difference in terms of the surface cleanness and thermal effect zone at the hole exit in Figure 5, meaning the advantage to drilling in water is not obvious. Moderate discussion on it needs to amend.
Reviewer 2 Report
The experimental research of millisecond laser trepanning in superalloy GH4220 with water-based assistance was carried out in this paper, and the effect of different pulse energy on the spatter, hole diameter, roundness, taper angle, sidewall morphology, and roughness, and the distribution of recast layer in air and with water-based assistance was studied. The author should consider the following question.
1. the explanation for the oxidation around the hole in Fig.4 is not believable. why does there almost no oxidation for the sample in the air?
Round 2
Reviewer 1 Report
Questions from the reviewer especially the submerged depth in the water and the orientated coloration of hole surrounding region have been well addressed in the revised version. Thank you.